# The Use of ESEM-EDX as an Innovative Tool to Analyze the Mineral Structure of Peri-Implant Human Bone

**DOI:** 10.3390/ma13071671

**Published:** 2020-04-03

**Authors:** Carlo Prati, Fausto Zamparini, Daniele Botticelli, Mauro Ferri, Daichi Yonezawa, Adriano Piattelli, Maria Giovanna Gandolfi

**Affiliations:** 1Endodontic Clinical Section, School of Dentistry, Department of Biomedical and Neuromotor Sciences, University of Bologna, 40125 Bologna, Italy; fausto.zamparini2@unibo.it; 2Laboratory of Biomaterials and Oral Pathology, School of Dentistry, Department of Biomedical and Neuromotor Sciences, University of Bologna, 40125 Bologna, Italy; mgiovanna.gandolfi@unibo.it; 3Ardec Academy, 47923 Rimini, Italy; daniele.botticelli@gmail.com; 4Corporación Universitária Rafael Núñez, Cartagena de Indias 130014, Colombia; medicina2000ctg@hotmail.com; 5Department of Applied Prosthodontics, Graduate School of Biomedical Sciences, Nagasaki University, Nagasaki 852-8102, Japan; yonezawadental@gmail.com; 6Department of Medical Oral and Biotechnological Sciences, University of Chieti Pescara, 66100 Chieti, Italy; adriano.piattelli@unich.it

**Keywords:** ESEM-EDX, Retrieved implants, Ca/P ratio, Ca/N ratio, P/N Ratio, Bone–implant interface

## Abstract

This study aimed to investigate the mineralization and chemical composition of the bone–implant interface and peri-implant tissues on human histological samples using an environmental scanning electron microscope as well as energy-dispersive x-ray spectroscopy (ESEM-EDX) as an innovative method. Eight unloaded implants with marginal bone tissue were retrieved after four months from eight patients and were histologically processed and analyzed. Histological samples were observed under optical microscopy (OM) to identify the microarchitecture of the sample and bone morphology. Then, all samples were observed under ESEM-EDX from the coronal to the most apical portion of the implant at 500x magnification. A region of interest with bone tissue of size 750 × 500 microns was selected to correspond to the first coronal and the last apical thread (ROI). EDX microanalysis was used to assess the elemental composition of the bone tissue along the thread interface and the ROI. Atomic percentages of Ca, P, N, and Ti, and the Ca/N, P/N and Ca/P ratios were measured in the ROI. Four major bone mineralization areas were identified based on the different chemical composition and ratios of the ROI. Area 1: A well-defined area with low Ca/N, P/N, and Ca/P was identified as low-density bone. Area 2: A defined area with higher Ca/N, P/N, and Ca/P, identified as new bone tissue, or bone remodeling areas. Area 3: A well-defined area with high Ca/N, /P/N, and Ca/P ratios, identified as bone tissue or bone chips. Area 4: An area with high Ca/N, P/N, and Ca/P ratios, which was identified as mature old cortical bone. Bone Area 2 was the most represented area along the bone–implant interface, while Bone Area 4 was identified only at sites approximately 1.5 mm from the interface. All areas were identified around implant biopsies, creating a mosaic-shaped distribution with well-defined borders. ESEM-EDX in combination with OM allowed to perform a microchemical analysis and offered new important information on the organic and inorganic content of the bone tissue around implants.

## 1. Introduction

Oral bone remodeling processes close to the implant threads have become a topic of particular interest in modern implantology. Stable peri-implant bone tissue is essential to achieving long-term results [1,2,3]. The peri-implant bone interface has been investigated on animal models, evidencing tight bone to implant contact with the implant and bone [4,5].

Bone is a dynamic tissue which is continuously modified in response to mechanical demands; these modifications lead to significant variations in the mineral and organic content [6]. These variations reflect the healing and the health conditions of bone [7]. 

An environmental scanning electron microscope (ESEM) is a repeatable investigation which is able to analyze mineralized tissue micromorphology without sample manipulation or deterioration [8,9], and may produce attractive results in the investigation of the peri-implant bone interface. Energy-dispersive x-ray spectroscopy (EDX) is useful for studying the composition of mineralized tissues or mineral (apatite) deposits and the mineralization degree of bone by calculating the element ratios (atomic or weight); furthermore, it makes it possible to detect elements that may have migrated from an implanted material into the bone tissue. 

The calcium-to-phosphorous (Ca/P) ratio has been used in several studies to assess the level of bone mineralization. Likewise, atomic calcium-to-nitrogen (Ca/N) and phosphorus-to-nitrogen (P/N) ratios have been used to investigate the degree of mineralization by evaluation of organic components in dentin and bone tissue [8,9,10,11,12].

To date, very few studies have analyzed the degree of mineralization at the peri-implant bone interface and around dental implants [13,14]. The present study aimed to analyze the mineralization degree along the bone–implant surface and in marginal bone tissue of a clinically stable retrieved human dental implant after four months from the unloaded placement with a novel technique based on the use of ESEM-EDX.

## 2. Materials and Methods 

The protocol of the study was approved by the Ethical Committee of the Corporación Universitária Rafael Núñez, Cartagena de Indias, Colombia.

Detailed information on Material and Methods and histological data have been tuned and reported by Gandolfi et al. in previous papers [8,9]. In this study, 16 healthy, nonsmoker volunteers received two mini implants (Sweden & Martina, Due Carrare, Padova, Italy, 5 mm height, 3.5 mm width). Implants were placed following a nonsubmerged procedure in the edentulous distal region of the mandible by a skilled operator. After local anesthesia, full-thickness flaps were raised and an osteotomy was performed with a series of drills of increasing diameter. Final alveolar socket preparation was 2.8 mm in width in the apical portion, 3.0 mm in width in the coronal portion, and 5.0 mm in depth. Implants were placed with the coronal margin flush to the bony crest (nonsubmerged placement). During the follow-up, five patients suffered from Chikungunya viral infections and were therefore excluded from the study [15]. After 4 months, mini implants were removed with a trephine bur. Biopsy specimen containing the implants were fixed in 10% buffered formalin immediately after retrieval. One additional patient was excluded as the obtained biopsies were damaged during processing [15]. A total of 10 unloaded implants were available for further analyses.

### 2.1. Histological Preparation

The specimens were first dehydrated in alcohol and then included in a glycol methacrylate resin (Technovit 7200 VLC; Kulzer, Wehrheim, Germany). Subsequently, they were polymerized and sectioned using a diamond steel disc along the major axis of the implants at approximately 150 mm and ground to about 30 microns. The sections were stained with acid fuchsin and toluidine blue. At the end of the procedure, eight ground sections from eight patients (five females, three males, mean age 54.3 ± 9.1 years) were available for both for Optical microscopy (OM) and ESEM-EDX Microanalysis.

### 2.2. OM and ESEM-EDX Microanalysis

The histological samples were observed under optical microscopy (OM) to identify the macrostructure of the sample and bone macromorphology.

Then, ESEM observation was preformed entirely, from the coronal to the most apical portion of the implant at 500x magnification. The specimens were placed directly on the ESEM stub and examined without any previous preparation (uncoated samples). Operative parameters were: low vacuum 100 Pascal, accelerating voltage of 20–25 kV, working distance 8.5 mm, and 133 eV resolution in Quadrant Back-Scattering Detector (QBSD) mode (0.5 wt% detection level, amplification time 100 μs, measuring time 60 s).

Of these, a region of interest (ROI) of 750 × 500 microns in size was selected to correspond with the first coronal and the last apical thread with bone tissue. Images at high magnification (5000x) were also performed to assess the quality of the interface between implant and bone.

EDX analyses were carried out at areas of approximately 30 × 30 microns, and qualitative and semiquantitative element (weight % and atomic %) content were investigated by applying the ZAF correction method, a procedure in which corrections for atomic number effect (Z), absorption (A), and fluorescence (F) are calculated separately [8].

For all the acquired spectra, the atomic Ca/N, P/N, and Ca/P ratios were calculated to evaluate the degree of mineralization, according to a previous study [9].

Different bone mineralization areas were detected and measured in um^2^ on each ROI using the Image J software (NIH software, Bethesda, MD, USA). Dark-light grayscale quantification was used to isolate and measure different gray values areas. For each ROI, an additional operator performed three measurements and the mean values were recorded in a spreadsheet. The percentage of each bone area was calculated on the total ROI. Analyses of distant bone areas were additionally performed to assess the mineralization of mature cortical bone.

### 2.3. Statistical Analysis

To compare the mineralization values of coronal versus apical ROI, two-way ANOVA followed by a Holm–sidak test (normality test *p* > 0.05, equal variance test *p* > 0.05) were performed. *P*-value was previously set at 0.05. Two-way ANOVA power was 0.667.

## 3. Results

A total of eight healthy human unloaded implants retrieved four months after placement were analyzed. Based on the tissues electron density grade detected by ESEM investigation (Figure 1 and Figure 2) and on the differences of Ca, P, and N atomic content detected by EDX, four different bone areas were identified (Table 1 and Table 2 and Figure 1 and Figure 2) and grouped as follows:

Area 1: Low mineralized area with major organic content. Very low Ca/N and very low P/N, defined as low mineralized bone/bone marrow. ESEM appeared as a dark grey area.

Area 2: Partially mineralized area, which was attributable to bone remodeling and new bone formation areas. Medium Ca/N and P/N showed moderate mineralization values.

Area 3: Higher mineralized areas with high Ca/N and very high P/N, defined as high mineralized bone/bone chips.

Area 4: Highly mineralized areas with a dense and homogeneous structure. High Ca/N, very high P/N, and Ca/P attributable to cortical bone. This area was evidenced at bone sites 1.0 mm from the investigated ROI and in cortical bone that was not in contact with the implant.

The areas were identified by the combined use of ESEM-EDX. Each area resulted in a different scale of grey, and the border of each was well-defined and not overlapped (Figure 2). Correspondences were observed between OM observation and ESEM-EDX analysis. ROI resulted in mosaic-shaped areas.

The OM and ESEM images from two histological samples included in the present study are reported in Figure 1 and Figure 2. ESEM-EDX revealed three different bone areas in the investigated ROI. Tissue along the interface is not uniformly mineralized and revealed different bone and bone marrow areas.

The overall mineralized area percentages of the investigated coronal and apical ROI are reported in Table 3. Analysis of bone areas over ROI revealed a high percentage of low mineralized areas (Bone Area 1) at the apical ROI (mean percentages values were 43.2 ± 18.1). A higher presence of mineralized bone (Bone Area 3) was detected at the coronal ROI (mean percentages were 26.5 ± 10.2). These changes, however, were not statistically different (*p* < 0.05). Only a small amount of Bone Area 4 was identified on the periphery of ROI.

The percentages of bone areas detected along the implant–bone interface and in contact with the implant surface are reported in Table 4.

Analysis along the implant interface revealed the marked presence of new bone (Area 2, mean percentages values were 54.2 ± 19.1) and low presence of old bone tissue (Area 3, mean values were 13.2 ± 12.8). Moderate percentages of low mineralized areas (Area 1) were identified (mean values were 34.2 ± 22.1). No statistically significant differences were observed between bone areas along coronal and apical threads (*p* > 0.05). No Bone Area 4 was present along the bone–implant interface.

Figure 3 reports the mature cortical bone tissue of one of the analyzed implants. An analysis of distant bone located at the periphery of the histology sample revealed highly mineralized areas (areas 3 and 4) with few marrow spaces.

The presence of a low electron dense organic layer between the implant and bone was detected in all the regions of interest (ROI); this layer ranged from between 3 and 18 µm. (Figure 4) 

## 4. Discussion

In this study, the analysis of human mini implants retrieved four months after placement was useful in the evaluation of the implant–bone interface and for the definition of mineralization areas around implants. This period is critical for the early osseointegration phases implants placed nonsubmerged, as demonstrated in a recent clinical study, where implant with same surface micromorphology but with larger diameter and lenght were placed using a similar protocol [16]. Few animal [4,5,17,18] and human studies [9] have used different ultrastructural techniques to obtain additional information on the microchemical composition of bone tissue around implants. 

Compared to optical microscopy, ESEM-EDX microanalysis allowed for the assessment and quantification of the presence of different bone types based on an elemental analysis of Ca, P, and N. In this way, four different mineralization areas were detected, considering the relative atomic Ca, P (from the inorganic bone components), and N content (from organic components). Atomic Ca/N and P/N ratios were used to analyze and to identify the mineralization level of the coronal and apical ROI and, as a control, on bone tissue at the periphery of the histological samples (approx. 1.5 mm from coronal ROI).

Low mineralized areas were clearly identified and defined as Bone Area 1. The element composition of Ca and P was lower (i.e., the lowest) than that observed in other ROI. Although no statistical significance was observed (*p* > 0.05), many Area 1 mineralizations were in contact with the implant surface, especially around the apical portion, and occupied approximately 30% of the ROI. Bone marrow low-density areas have been described in histological studies and are present in many animal model investigations [14,18,19].

Bone Area 2 was identified at ESEM-EDX by high Ca, P and Ca/P and lower Ca/N and P/N atomic ratios. At OM investigation, this area showed a more complex architecture and high vascular presence; it occupied approximately 35% of the ROI. These structures were also characterized by the presence of osteocytes lacunae and osteons, which indicate physiological bone remodeling [15,20]. 

Bone Area 3 showed high levels of Ca and P and low N, with the result of lower Ca/N and P/N atomic ratios. This area occupied approximately 20–30% of the ROI. It was more evident at the cortical level and 200–500 microns from the implant surface. The element composition suggests a mature bone free from any resorption process. 

Old bone chips from previous surgical drilling have been observed and described in several animal [18,19,20,21] or human histological studies, in particular, with high torque values or when the alveolar sites are underprepared [22]. They were included in the bone type Area 3.

The effects of bone chips at the peri-implant interface is controversial in the literature. Some studies suggest that these fragments may support bone formation directly on the implant surface [17,22], while others suggest that bone healing processes require additional time, as bone chips have to be firstly resorbed to leave space for new bone [22]. In the present study, bone chips at the interface appear to be perfectly integrated into the new bone, possibly corroborating the first hypothesis. 

Mature cortical bone was defined as Bone Area 4, which was characterized by slightly higher Ca/N and P/N ratios compared to Bone Area 3. This type of bone was detected in the periphery of ROI and far from the implant surface (approx. 1–1.5 mm from ROI). No traces of Bone Area 4 were detected at the thread interface.

Bone Areas 1, 2, and 3 were identified around implant biopsies creating a multicell system that, in the 2D section of the histology, appears as a mosaic-shaped distribution with well-defined borders.

After additional time, Bone Area 2 will likely turn into a more mineralized Area 3, and after years, an Area 4. In contrast, Bone Area 1 will remain relatively stable as a cell reservoir.

This study shows that the peri-implant bone interface is a heterogeneous and dynamic zone consisting of mineralized, partially mineralized, and unmineralized areas, in accordance with recent studies from Shah et al. [7,14]. 

The human peri-implant bone interface at four months from implant placement should be interpreted as a volumetric 3D wide zone where tissue modification and rearrangements take place, as demonstrated by the presence of chambers of Bone Area 1 with low concentrations of Ca and P, delimitated by Bone Areas 2 and 3 which are more mineralized and denser. This network is probably richer in mesenchymal cells than mature old bone (Area 4). The short time after implant placement (four months) may explain the presence of Bone Area 1, which is just smaller in biopsies (from retrieved implants) performed longer after insertion [9]. 

A study by Gandolfi et al. analyzed nine implants retrieved for mechanical complications, revealing a highly mineralized and homogeneous implant–bone interface with limited low mineralized areas after a long loading period (>10 years). However, unloaded implants retrieved two months after insertion demonstrated large percentages of low-mineralized bone tissue, some bone chips, as well as new bone formation at the thread interface, in line with the present study [9]. The low Ca concentration of Bone Area 1 suggests its role as a cell reservoir and a space for new collagen and bone formation. 

Limitations of the present study may be the low number of implant biopsies analyzed. A study taking into consideration a higher number of histological samples is needed to further confirm the obtained results.

In conclusion, ESEM-EDX in combination with OM allowed us to perform structural and microchemical analyses, offering important new information on the organic and inorganic content of bone in a critical phase of transformation.

## Figures and Tables

**Figure 1 materials-13-01671-f001:**
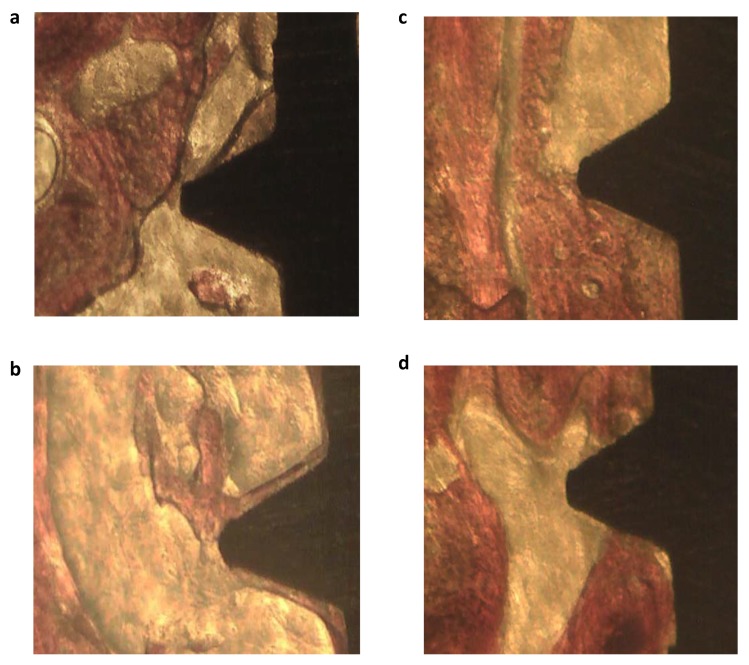
OM analysis of two selected coronal (**a**,**c**) and apical (**b**,**d**) ROI from two implant specimens at 200x magnifications. The apical and coronal side of the implant specimens revealed differences in their composition, and the apical side of both specimens revealed a higher presence of bone marrow areas. In contrast, coronal bone revealed a higher presence of remodeling areas (as evidenced by osteon structures).

**Figure 2 materials-13-01671-f002:**
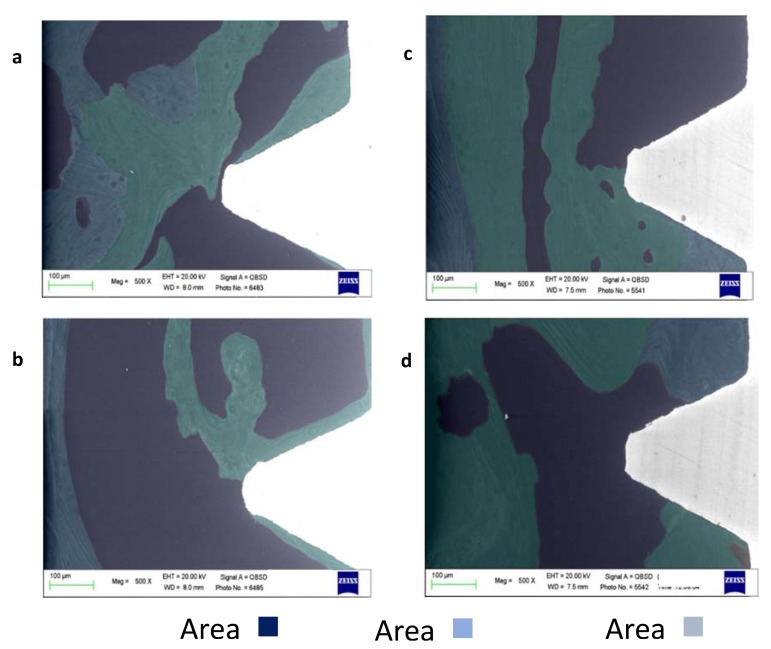
ESEM/EDX microanalysis of selected coronal and apical ROI of two different implants included in the study. (**a**) Coronal thread. The analysis revealed a heterogeneous bone tissue. The coronal side of the thread was in contact with moderately electron-dense bone (Area 2), while the apical side revealed highly electron-dense tissue (Area 3). Osteocytes lacunae and bone lamellae were concentrically arranged, partially in contact with the apical side of the thread. Remote bone followed a similar thread, with the presence of high mineralized bone structures (Area 3), in close contact with moderate mineralized bone (Area 2), indicating ongoing bone remodeling at the moment of the sample retrieval. (**b**) Microanalysis of the apical region revealed moderately electron-dense bone in contact with the implant thread. In this area, very low mineralized tissue was markedly present (Area 1), indicating bone marrow areas and suggesting an incomplete bone remodeling process. (**c**) Investigation of the coronal ROI revealed irregular electron-dense areas. Moderately and highly mineralized areas (Area 2 and 3) were identified on the apical side of the thread, while very low mineralized bone tissue was evident at the bone interface of the coronal side of the thread (Area 1). Bone that was distant from the implant interface revealed a heterogeneous structure (also evidencing numerous bone niches), mostly composed of Area 2 bone, suggesting new bone or bone remodeling at this site. (**d**) The apical ROI revealed the marked presence of moderately electron-dense bone tissue (Area 2) in close contact with the implant thread. Very low mineralized bone tissue (Area 1) was identified at the top of the implant thread, suggesting incomplete bone remodeling in this area. Highly mineralized bone (Area 3) was found in limited areas in contact with the implant thread, and on bone sites at a distance of approximately 500 µm.

**Figure 3 materials-13-01671-f003:**
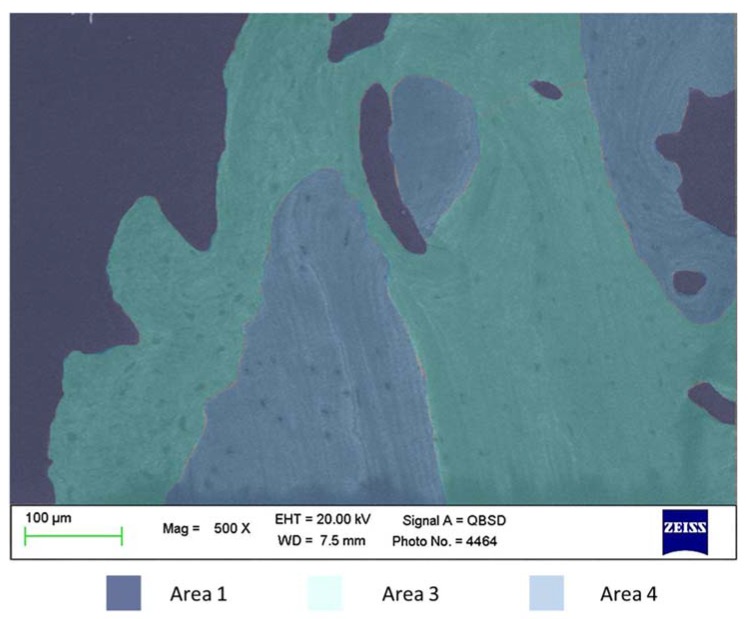
ESEM-EDX of bone areas at 1.5 mm from the coronal ROI. The presence of mineralized Bone Areas 3 and 4 was detected at this site. Bone Area 4 revealed higher Ca/N and P/N, but similar Ca/P ratios when compared to Area 3.

**Figure 4 materials-13-01671-f004:**
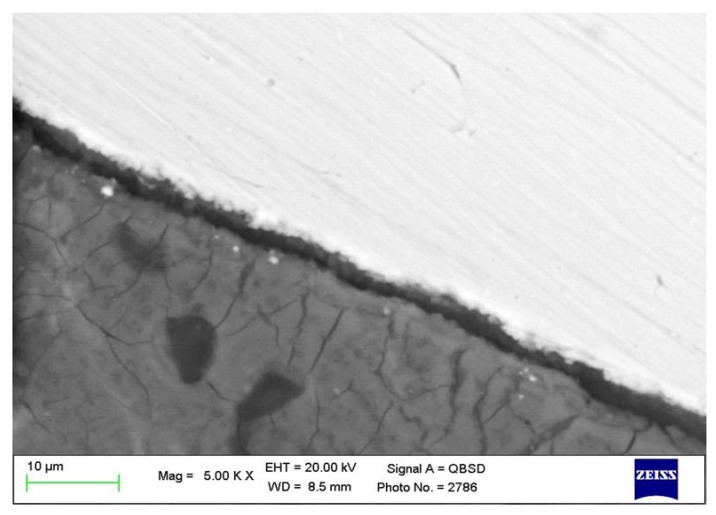
ESEM microanalysis at 5000x magnifications, revealing a low electron-dense layer between the implant and the bone tissue. This layer was detected along the bone–implant interface, with a size ranging from 3 to 18 µm.

**Table 1 materials-13-01671-t001:** Atomic values of Ca, P, and N in the analyzed areas of bone.

Mineralization Areas	Ca	P	N
Bone Area 1 low mineralized bone (bone marrow)	Very low 0.7–1.20	Very low 0.90–1.10	High 13.0–15.0
Bone Area 2 Medium mineralized bone (bone remodeling)	Moderate 1.25–1.75	Moderate 1.10–1.50	Moderate 11.0–12.0
Bone Area 3 high mineralized area (mature old bone/bone chips)	High 1.75–3.00	High 1.50–2.00	Low 8.0–11.0
Bone Area 4 high mineralized area (cortical bone)	Very High 2.75–5.00	Very High 2.00–3.00	Moderate 11.0–12.0

Four different areas of bone were identified based on Ca/N, P/N, and Ca/P and classified as bone marrow spaces (Area 1: characterized by low inorganic and high organic content), remodeling bone (Area 2: moderate inorganic and organic content) and mature old bone and bone chips (Area 3: high inorganic and low organic content). Bone Area 4, similar to Area 3 (but with a higher N content) was identified at sites distant from the investigated ROI and was attributable to mature cortical bone tissue.

**Table 2 materials-13-01671-t002:** Range of EDX atomic ratio of Ca/N P/N and Ca/P of the analyzed areas of bone.

Mineralization Areas	Ca/N	P/N	Ca/P
Bone Area 1 low mineralized bone (bone marrow)	Very low 0.01–0.08	Very low 0.02–0.08	Low 0.75–1.20
Bone Area 2 Medium mineralized bone (bone remodeling)	Moderate 0.11–0.16	Moderate 0.10–0.20	Moderate 1.30–1.50
Bone Area 3 high mineralized area (mature old bone /bone chips)	High 0.18–0.25	High 0.15–0.25	High 1.30–1.80
Bone Area 4 high mineralized area (cortical bone) apatite	High 0.18–0.25	High 0.15–0.20	High 1.33–2.00

**Table 3 materials-13-01671-t003:** Type of bone areas measured at ROI (Mean % ± SD) n = 8 implants. No statistically significant differences were reported among coronal and apical ROI (*p* < 0.05)**.**

ROI	Bone Area 1 low Mineralized Bone (Bone Marrow)	Bone Area 2 Medium Mineralized Bone (New Bone)	Bone Area 3 High Mineralized Area (Mature Old Bone/Bone Chips)
Coronal ROI	34.9 ± 14.7	39.9 ± 12.1	26.5 ± 10.2
Apical ROI	43.2 ± 18.1	36.8 ± 12.9	19.8 ± 9.7

**Table 4 materials-13-01671-t004:** Type of bone area detected at the implant–bone interface detected along the coronal and apical portion, calculated as percentages (Mean ± SD). N = 8 implants. No statistically significant differences were reported among coronal and apical ROI (*p* < 0.05).

Thread	Bone Area 1 Low Mineralized Bone (Bone Marrow)	Bone Area 2 Medium Mineralized Bone (Bone Remodeling)	Bone Area 3 Highly Mineralized Area (Mature Old Bone/Bone Chips)
Coronal	34.2 ± 22.1	54.2 ± 19.1	13.2 ± 12.8
Apical	32.4 ± 20.1	50.2 ± 14.7	15.1 ± 12.1

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
