# Peer review of "The Use of ESEM-EDX as an Innovative Tool to Analyze the Mineral Structure of Peri-Implant Human Bone"

_materials, 2020, doi:10.3390/ma13071671_

Round 1

Reviewer 1 Report

The goal of this study was to investigate the mineralization area and further understand the microchemistry of the bone at the perimplant site of human bone with ESEM-EDX analyses. 

The overall conclusion that four distinct mineralization sites are observed is consistent with the well-recognized dynamics of bone remodeling, but the paper presents further detailed assessments of the nature of the mineralization in the microenvironment of an implant. 

While this study does add to our knowledge about mineralization at a perimplant site with their use of ESEM and EDX, lack of details of the subjects as well as the nature of their data collections, appear to limit the information that a reader could obtained from the paper as presented. 

The paper could provide more useful data that could be extrapolated to clinical evaluations if more details are given regarding the human subjects in which the implants were placed.  The basic demographics of the human subjects such as their age, sex, smoking habits, general health, etc.  would be important in order to extrapolate to what would be expected in a clinical situation.

 In the methods section, it is stated that there were 2 implants per subject and one was randomly selected to be loaded with a single crown while the other was left unloaded.   In the results, however, the loaded versus unloaded sites do not appear to be differentiated with respect to the mineralization sites. The effects of loading should be more thoroughly addressed.

It is mentioned that 5 subjects were used in the study but the manner in which the data in Tables 2 and 3 are expressed with respect to a single or combined subjects needs to be elaborated.  It is presumed the data are in mean+/- std dev, but this needs to be stated in a legend to the Tables.   

It is stated that more information about the methods can be found in cited references from the authors, but it is difficult for readers to find these papers and extract the basic information that should be presented here.

Author Response

To the editor board to nanomaterials:

We modified the text according with referees comment. All the modifications have been highlighted in yellow in the text.

REFEREE 1

The goal of this study was to investigate the mineralization area and further understand the microchemistry of the bone at the perimplant site of human bone with ESEM-EDX analyses.

The overall conclusion that four distinct mineralization sites are observed is consistent with the well-recognized dynamics of bone remodeling, but the paper presents further detailed assessments of the nature of the mineralization in the microenvironment of an implant.

While this study does add to our knowledge about mineralization at a perimplant site with their use of ESEM and EDX, lack of details of the subjects as well as the nature of their data collections, appear to limit the information that a reader could obtained from the paper as presented.

 The paper could provide more useful data that could be extrapolated to clinical evaluations if more details are given regarding the human subjects in which the implants were placed.  The basic demographics of the human subjects such as their age, sex, smoking habits, general health, etc.  would be important in order to extrapolate to what would be expected in a clinical situation.

We thank the referee for the suggestion. All patients included in the present study were healthy and non smokers. Implant placement was performed in the posterior region of the maxilla. This information has been added in the text.

 In the methods section, it is stated that there were 2 implants per subject and one was randomly selected to be loaded with a single crown while the other was left unloaded.   In the results, however, the loaded versus unloaded sites do not appear to be differentiated with respect to the mineralization sites. The effects of loading should be more thoroughly addressed.

We agree with the referee that loading could be a significative factor to be investigated. The analysed samples were gathered from a study taking account analysis of loaded versus unloaded samples (Yonezawa et al 2018). However, in the present investigation, only unloaded implants have been evaluated, as the principal aim was to characterize bone mineralization and microchemistry at the implant interface using OM and ESEM-EDX. For these reasons, the implant location, implant placement and surgery type were standardized. The effect of loading procedures will be assessed in further ongoing studies.  

Yonezawa D, Piattelli A, Favero R, Ferri M, Iezzi G, Botticelli D. Bone Healing at Functionally Loaded and Unloaded Screw-Shaped Implants Supporting Single Crowns: A Histomorphometric Study in Humans. Int J Oral Maxillofac Implants. 2018;33:181-187.

It is mentioned that 5 subjects were used in the study but the manner in which the data in Tables 2 and 3 are expressed with respect to a single or combined subjects needs to be elaborated.  It is presumed the data are in mean+/- std dev, but this needs to be stated in a legend to the Tables.  

Table 2 and 3 report mean and standard deviation of single unloaded implants (one for each patient). Authors would like to clarify that the analysis now considered a total of 8 unloaded implants, aiming to have a larger sample to evaluate. This has been added in the methodology section. Legend in the table has been modified, stating the number of implants analyzed and that values were mean and SD.

It is stated that more information about the methods can be found in cited references from the authors, but it is difficult for readers to find these papers and extract the basic information that should be presented here.

We thank the referee for the indication. More details on surgical approach, biopsies gathering, patient drop out and available specimens has been added in the methodology section.

Reviewer 2 Report

The authors performed a standardless (semi-quantitative) analysis referring to a protocol previously described in refs 8 and 9. I have no access to ref 9 but ref 8 is not appropriate for a full protocol as the information presented there is minimal. Can you please describe in more detail how you dealt with the matrix effects for the light elements? This might also explain, apart from the heterogeneity of the bone tissue, the large SDs in tables 2 and 3.

In lines 48 you refer to SEM as non-destructive technique. Then you measured stained ground sections.

There is a discrepancy between the 16 volunteers and the 5 samples subsequently selected. Why?

It is not clear if you performed element ratio calculations based on % atomic or % weight content. Was there any difference? What would be the expected difference in the context of this work? Which ratio is presented in the tables?

Is there any specific reason for the post-hoc Student-Newman-Keuls test? Please note that this test is inferior to Holm-Sidak test and it cannot produce 95% confidence intervals for the calculated differences. Moreover, statistical analysis was not properly discussed. I couldn’t find any hint on any differences but in line 155.

Ordinary two-way ANOVA requires normal data. Have you tested for normality?

In results, dark/light greyscale needs intensity quantification to better specify the areas 1-4. This can be done by Image-J and then possibly correlated to the EDX measurements.

In Discussion, the ultra small paragraphs do not convey the significance of the work. Please try to assemble paragraphs with a normal text flow. Furthermore, try to include some statistics to your analyses.

Author Response

REFEREE 2

The authors performed a standardless (semi-quantitative) analysis referring to a protocol previously described in refs 8 and 9. I have no access to ref 9 but ref 8 is not appropriate for a full protocol as the information presented there is minimal. Can you please describe in more detail how you dealt with the matrix effects for the light elements? This might also explain, apart from the heterogeneity of the bone tissue, the large SDs in tables 2 and 3.

We thank the referee for the indication. Reference 8 has been replaced with another study, were ESEM EDX was carried out with a similar protocol. In order to deal with the matrix effects for light elements, ZAF correction has been applied to the acquired spectra.

In lines 48 you refer to SEM as non-destructive technique. Then you measured stained ground sections.

The authors intended that ESEM analysis is repeatable and non destructive as the sample can be re-examined and used also for further analyses. “Non-destructive” has been removed, according referee suggestion.

There is a discrepancy between the 16 volunteers and the 5 samples subsequently selected. Why?

Paragraph has been rewritten to be clearer. Out of 10 volunteers, 2 were excluded according with a recent published study, Yonezawa et al 2018 as not evaluable. We implemented the analysed biopsies and now 8 unloaded specimens are analysed 8 both available for OM and ESEM investigation.

Yonezawa D, Piattelli A, Favero R, Ferri M, Iezzi G, Botticelli D. Bone Healing at Functionally Loaded and Unloaded Screw-Shaped Implants Supporting Single Crowns: A Histomorphometric Study in Humans. Int J Oral Maxillofac Implants. 2018;33:181-187.

It is not clear if you performed element ratio calculations based on % atomic or % weight content. Was there any difference? What would be the expected difference in the context of this work? Which ratio is presented in the tables?

Ratio calculations were performed on the basis of atomic % content. Ratios calculated on the basis of weight% revealed a higher variation, attributable to element with higher z value (i.e. Ca and P).

Overestimation of Ca and P in bone tissues may be expected when analyzing weight ratios also according with a study of Akesson et al. 

Akesson K, Grynpas MD, Hancock RG, Odselius R, Obrant KJ. Energy-dispersive X-ray microanalysis of the bone mineral content in human trabecular bone: a comparison with ICPES and neutron activation analysis. Calcif Tissue Int. 1994 Sep;55(3):236-9.

Moreover, atomic % ratios calculation has been used in the mineralization assessment in several published studies on mineralized tissues, such as bone tissue (Kourkoumelis 2012) peri-implant bone (Shah et al. 2014) or dentin tissue (Eliades et al. 2013).

Shah FA, Nilson B, Brånemark R, Thomsen P, Palmquist A. The bone-implant interface nanoscale analysis of clinically retrieved dental implants. Nanomedicine. 2014;10:1729-1737.

Kourkoumelis N, Balatsoukas I, Tzaphlidou M. Ca/P concentration ratio at different sites of normal and osteoporotic rabbit bones evaluated by Auger and energy dispersive X-ray spectroscopy. J Biol Phys 2012;38:279-291

Eliades G, Mantzourani M, Labella R, Mutti B, Sharma D. Interactions of dentine desensitisers with human dentine: morphology and composition. J Dent 2013;41:28-39.

Is there any specific reason for the post-hoc Student-Newman-Keuls test? Please note that this test is inferior to Holm-Sidak test and it cannot produce 95% confidence intervals for the calculated differences. Moreover, statistical analysis was not properly discussed. I couldn’t find any hint on any differences but in line 155. Ordinary two-way ANOVA requires normal data. Have you tested for normality?

We thank the referee for the indication. Holm-sidak test has been now performed instead of Student-Newman-Keuls test. Statistical analyses has been performed using Sygmaplot 12 software, both normality test and equal variance tested passed (p value was > 0.05) for the statistical analyses of table 2 and 3.

In results, dark/light greyscale needs intensity quantification to better specify the areas 1-4. This can be done by Image-J and then possibly correlated to the EDX measurements.

Image J dark light/grey scale quantification has been used to calibrate the bone areas measurement before the assessment, to better identify the different mineralization area. This information has been added in the methodology section.

In Discussion, the ultra-small paragraphs do not convey the significance of the work. Please try to assemble paragraphs with a normal text flow. Furthermore, try to include some statistics to your analyses.

Paragraphs have been reassembled, following referee suggestion. In addition, statistical analysis has been added.

Reviewer 3 Report

This is an interesting piece of work, with clear impact on implant development and understanding on physiological response post-implantation. 

The authors are presenting a less explored technique in this field (ESEM-EDX) to extract valuable conclusions and characterise bone presence around these dental implants.

The major issues of this manuscript are related to the quality or clearness of the writing, as well as to options in the applied methodology.

Abstract, line 18, "interface bone"? Maybe bone-implant interface?

Line 26, "after 4 months unloaded", more context is needed to clarify the concept of "unloaded" under these circumstances.

The abstract is very important for the first contact with the paper, so please make sure that you revise it and address these little comments. In addition, please discuss the results instead of jumping directly to the conclusions that the applied techniques are "important".

Introduction, line 60, "for it is very difficult", please revise the English and clarify the actual difficulties in characterising such objects of study.

Line 62, "clinically stable retrieved human dental implant at 4 months unloaded", very confusing statement and lacking context.

Materials and methods is the critical section in this manuscript. 16 volunteers are mentioned in line 67, but then only data from 5 patients is considered, line 77? Why? 

The protocol was approved by a Colombian institution, but no Colombian affiliations are stated in the authors list? How is this so? Also, more information about the work performed at reference [19] is needed to provide context to the readers. 

Why 2 + 2 months? Why was this period of time selected for this study and why weren't multiple time points analysed?

Line 69, "non-submerged fashion"? Please revise.

Line 70, please clarify the loading/unloading situation. Why are the so-called "loaded implants" not analysed here?

Procedures described in lines 89-93 are very confusing. Please revise and clarify the "blinded assessor" and "data were calculated as percentages of area".

Results seem to be aligned with the methods, as well as the discussion seems to be in accordance with the results. However, this paper needs to be revised and its limitations need to be further considered before publication. In lines 209-210, authors mention that these results were probably expectable, so where is the actual step forward provided by this paper? A wider study with these techniques could really add to the literature, so authors can also take this chance to enrich their research.

Author Response

REFEREE 3

This is an interesting piece of work, with clear impact on implant development and understanding on physiological response post-implantation.

The authors are presenting a less explored technique in this field (ESEM-EDX) to extract valuable conclusions and chracterise bone presence around these dental implants.

The major issues of this manuscript are related to the quality or clearness of the writing, as well as to options in the applied methodology.

Abstract, line 18, "interface bone"? Maybe bone-implant interface?

We thank the referee, interface bone has been modified

Line 26, "after 4 months unloaded", more context is needed to clarify the concept of "unloaded" under these circumstances.

Abstract has been modified following the referee suggestions

The abstract is very important for the first contact with the paper, so please make sure that you revise it and address these little comments. In addition, please discuss the results instead of jumping directly to the conclusions that the applied techniques are "important".

Abstract has been modified following the referee suggestions

Introduction, line 60, "for it is very difficult", please revise the English and clarify the actual difficulties in characterizing such objects of study.

Introduction section has been modified following the referee suggestions.

Line 62, "clinically stable retrieved human dental implant at 4 months unloaded", very confusing statement and lacking context.

Introduction section has been modified following the referee suggestions.

Materials and methods is the critical section in this manuscript. 16 volunteers are mentioned in line 67, but then only data from 5 patients is considered, line 77? Why?

Paragraph has been rewritten according with referees’ indications

The protocol was approved by a Colombian institution, but no Colombian affiliations are stated in the authors list? How is this so? Also, more information about the work performed at reference [19] is needed to provide context to the readers.

More information of surgical approach, implant placement has been provided. Moreover, surgical procedures were performed in a Colombian University by a skilled operator now included as co-author in the present study.

Why 2 + 2 months? Why was this period of time selected for this study and why weren't multiple time points analysed?

Paragraph has been rewritten according with referees’ indications

Line 69, "non-submerged fashion"? Please revise.

The term has been revised.

Line 70, please clarify the loading/unloading situation. Why are the so-called "loaded implants" not analysed here?

We agree with the referee that loading could be a significative factor to be investigated. However, in the present investigation, only unloaded implants have been evaluated, as the principal aim was to characterize bone mineralization and microchemistry at the implant interface using OM and ESEM-EDX. For these reasons, the implant location, implant placement and surgery type were standardized. The effect of loading procedures will be assessed in further ongoing studies. 

Procedures described in lines 89-93 are very confusing. Please revise and clarify the "blinded assessor" and "data were calculated as percentages of area".

Sentence has been revised.

Results seem to be aligned with the methods, as well as the discussion seems to be in accordance with the results. However, this paper needs to be revised and its limitations need to be further considered before publication. In lines 209-210, authors mention that these results were probably expectable, so where is the actual step forward provided by this paper? A wider study with these techniques could really add to the literature, so authors can also take this chance to enrich their research.

Discussion section has been modified following the referee suggestions.

Round 2

Reviewer 2 Report

Just a minor correction: "We thank the referee for the indication. Holm-sidak test has been now performed instead of Student-Newman-Keuls test."

L113 needs to be modified to include this info (probably this can be done in the proofs) 

Author Response

REFEREE 2

Just a minor correction: "We thank the referee for the indication. Holm-sidak test has been now performed instead of Student-Newman-Keuls test."

L113 needs to be modified to include this info (probably this can be done in the proofs)

Answer: We thank the referee, we modified the section.

Reviewer 3 Report

This manuscript looks better now, but the English language still needs to be revised. Some of the sentences included in this iteration are as confusing as the previous ones.

The format of the discussion is also not adequate.

If the authors can present an improved version in terms of format and language, this manuscript will be fine for publication.

Please also note that the identification of the authors should be in accordance with the publication guidelines, so name - surname. It looks like the current format is surname - name.

Author Response

REFEREE 3

This manuscript looks better now, but the English language still needs to be revised. Some of the sentences included in this iteration are as confusing as the previous ones.

The format of the discussion is also not adequate.

If the authors can present an improved version in terms of format and language, this manuscript will be fine for publication.

Answer: Format and English language has been revised as referee indicated.

Please also note that the identification of the authors should be in accordance with the publication guidelines, so name - surname. It looks like the current format is surname - name.

Answer: Thank you, the correct name and surname format has been now used.